# Characterization of Antimicrobial Resistance in *Escherichia coli* Isolated from Diarrheic and Healthy Weaned Pigs in Catalonia

**DOI:** 10.3390/ani14030487

**Published:** 2024-02-01

**Authors:** Biel Garcias, Marga Martin, Laila Darwich

**Affiliations:** Department Sanitat i Anatomia Animals, Veterinary School, Universitat Autonoma de Barcelona, 08193 Cerdanyola del Valles, Spain; biel.garcias@uab.cat

**Keywords:** *Escherichia coli*, antimicrobial resistance, postweaning diarrhea

## Abstract

**Simple Summary:**

Postweaning diarrhea (PWD) is a significant concern in the swine industry, causing substantial economic losses due to decreased growth rates, increased mortality, and the extensive use of antimicrobials. Certain *Escherichia coli* (*E. coli*) pathovars are frequently implicated in PWD cases among piglets. Regular surveillance and testing of *E. coli* susceptibility to different antimicrobials is essential. This helps veterinarians and farmers select appropriate treatments, avoid unnecessary antibiotic use, and prevent the further spread of antimicrobial resistance (AMR). Herein, we tested the susceptibility to fourteen antimicrobials of 251 *E. coli* strains isolated from fecal samples of diarrheic (n = 148) and apparently healthy piglets (n = 103) in farms in Catalonia. The results of this study showed that 41.4% of the *E. coli* were multidrug-resistant (MDR) strains, presenting high resistance to conventional veterinary antimicrobials such as erythromycin, amoxicillin, streptomycin, tetracycline, lincospectin, sulfamethoxazole/trimethoprim, and florfenicol. A special concern was also observed for human last-resort antimicrobials, like quinolones and colistin. Moreover, pigs suffering from diarrhea had a higher frequency of infection with MDR strains than the healthy ones. To reduce the incidence and impact of PWD in pig herds, optimization of antimicrobial therapies had to be implemented with other strategies, such as good hygiene practices, optimizing nutrition, managing stress levels, and employing proper vaccination protocols.

**Abstract:**

Postweaning diarrhea (PWD) is a multifactorial concern in the swine industry that leads to high antibiotic consumption, usually without testing susceptibility, increasing the risk of the selection of *Escherichia coli*-resistant strains. In this study, 251 *E. coli* strains isolated from fecal samples of diarrheic (n = 148) and apparently healthy piglets (n = 103) in farms in Catalonia were tested against their susceptibility to fourteen different antimicrobials. The phenotypic antimicrobial resistance (AMR) testing revealed high levels of AMR, with 41.4% of the isolates presenting a multidrug-resistant (MDR) profile. More specifically, resistance to class D (prudence) antimicrobials such as erythromycin (99.6%), amoxicillin (95.2%), streptomycin (91.6%), tetracycline (88.8%), lincospectin (64.5%), and sulfamethoxazole/trimethoprim (60%) was very high, as well as to class C (caution) antimicrobials such as florfenicol (45%). A special concern was observed for antimicrobial category B (restrict), like quinolones and colistin, that both presented a high rate of resistance. Colistin use was substantially reduced in Spain, but resistance is still present in weaned pigs, presenting a MIC90 of 4 μg/mL. This suggests that reducing antibiotic use is not enough to eliminate this AMR. Finally, it was found that piglets suffering diarrhea were more commonly carriers of MDR strains than the healthy ones (49.3% vs. 35%, *p* = 0.031). Therefore, given the high rates of resistance to the most commonly used antimicrobials, especially in diseased pigs, a new non-antibiotic-based approach should be implemented for the management of PWD.

## 1. Introduction

In natural conditions, piglets are weaned in a gradual way at 12 weeks of life [1]. However, intensive productive conditions in the swine industry are quite different since animals are early-weaned at 3–4 weeks, when their intestines and immune system have not matured yet. This circumstance represents a stressful situation for the young animals, which leads them to anorexia that can last for even 48 h, with associated intestinal inflammation [2]. Moreover, stress generates an immunosuppressive state, which makes pigs susceptible to infectious agents [3]. Finally, the sudden change from a liquid to a solid diet leads to changes in the microbiota composition and diversity, which, although not fully understood, make piglets more susceptible to PWD [4].

Adding more complexity to the problem, postweaning diarrhea (PWD) has a multifactorial etiology. *Escherichia coli* (*E. coli*) is the main bacteria implicated in the process, with different virulence factors implicated, such as fimbriae F4 and F18, thermostable (STa, STb) and thermolabile (LT) toxins, verotoxins (VT1 and VT2), or the gene that encodes for intimin *eae* [5]. However, previous research on the role of *E. coli* as a main pathogen in the development of diarrhea reported nonconclusive results [6]. Viral infections are also commonly related to intestinal disorders, especially those caused by Rotavirus A, B, and C [7], as well as coronaviruses such as the porcine epidemic diarrhea virus or transmissible gastroenteritis virus. All these viruses have been identified as primary enteric pathogens [6,8,9]. On the other hand, non-infectious causes like nutrition, bad husbandry practices, or any stressful situation could lead to the development of PWD [10]. Furthermore, various factors can coexist simultaneously, making PWD a complex and multifactorial concern. In consequence, identifying a precise cause or origin can pose significant challenges. Nevertheless, young animals find themselves in a particularly vulnerable state, prompting the implementation of antimicrobial treatments, even in cases where a bacterial cause has not been definitively confirmed as the etiological agent.

As already expected by Alexander Fleming, antimicrobial use over the years has led to the apparition of antimicrobial resistance (AMR) bacteria. This has been specifically proven in swine production [11,12], and principally, it has been shown that AMR levels were significantly high at the weaning phase [13]. Reduced effectiveness of antimicrobial agents may lead to higher mortality rates from bacterial infections, posing a threat to animal well-being and causing significant economic losses. Moreover, AMR has become an important cause of death in humans [14]. But, although an epidemiological link has been found [15], livestock contribution to human AMR remains unclear since this relationship seems not to be present at the genomic level [16,17]. However, transmission of AMR genes via plasmids from livestock to humans has been proven [18], and it has been shown that people in close contact with farms have a higher risk of certain AMR bacteria, such as methicillin-resistant *Staphylococcus aureus* [19,20], and those bacteria are carriers of more AMR genes [21]. Therefore, AMR bacteria in livestock should be considered a public health threat, especially for farm and slaughterhouse workers.

The European Medicines Agency (EMA) categorizes antimicrobials used in veterinary medicine into different categories based on their importance, intended use, and impact on antimicrobial resistance [22]. These categories are designed to guide veterinarians and stakeholders in the responsible use of these medications, determining which antimicrobials veterinarians can use as first-line treatments and those that should be avoided or restricted. Category A (avoid) includes antimicrobials that are deemed critically important for human medicine and are not authorized for veterinary use (i.e., carbapenems). Category B (restrict) includes antimicrobials considered important for human medicine and should be used in animals with great caution and only when no suitable alternatives are available and when they are based on antimicrobial susceptibility testing (i.e., 3rd and 4th generation cephalosporins, quinolones, and polymyxins). In categories C (caution) or D (prudence), antimicrobials like the 1st generation of cephalosporins, aminoglycosides, chloramphenicol, penicillin/β-lactam inhibitors, tetracyclines, and trimethoprim/sulfonamides are included. These antimicrobials are considered acceptable and authorized for veterinary use as initial treatments for bacterial infections. It is important to follow these guidelines to prevent antibiotic resistance and ensure the responsible use of antimicrobials in veterinary medicine.

The study of AMR in livestock is indeed crucial. Techniques like whole-genome sequencing (WGS) of bacterial genomes and metagenomics play a significant role in identifying and understanding AMR determinants and transmission routes in various bacterial populations [23,24]. However, despite these advanced techniques, there are instances, as in the case of porcine *E. coli*, where phenotypical susceptibility analyses remain essential. This is especially important in PWD, where animals often require prompt antibiotic treatment before the availability of bacteriological diagnosis and antibiotic susceptibility test results. Therefore, field studies are crucial in creating adequate guidelines to help veterinary practitioners use antimicrobials in ways that prioritize animal health while minimizing the risk of antimicrobial resistance development.

In the present study, *E. coli* strains isolated from farms experiencing PWD in Catalonia, an area known for high pig production, were analyzed to understand the antibiotic susceptibility patterns. Moreover, this study aimed to investigate whether there are distinct antimicrobial susceptibility phenotype patterns in *E. coli* strains associated with PWD in diarrheic piglets compared to those found in healthy animals within the same environment.

## 2. Materials and Methods

### 2.1. Bacterial Identification

*E. coli* strains were isolated from 17 pig farms (14 farms with active and recurrent PWD outbreaks and 3 farms with sporadic PWD outbreaks in the last 12 months) between February 2020 and December 2021. All the studied farms were distributed in Catalonia (NE of Spain), the Spanish region with the greatest number of intensive production farms and one of the highest pig-density regions in Europe. A total of 251 fecal samples were collected from 3- to 5-week-old piglets; 148 out of them were diarrheic animals, and 103 were healthy pen mates that were considered healthy controls.

One gram of fresh stool samples was obtained directly from the animals using sterile rectal swabs. Swab samples were transported on Amies transport medium (Deltalab, Rubí, Spain) and submitted for diagnostic testing to the Veterinary Infectious Diseases Diagnostic Laboratory of the Autonomous University of Barcelona (UAB) (Bellaterra, Spain). All samples were cultured in Columbia blood agar (BD GmBh, Heidelberg, Germany) and MacConkey agar (Oxoid, Basingstoke, UK) and incubated for 24 h at 37 °C. Suspected colonies were identified using conventional biochemical tests (oxidase, catalase, TSI, SIM, urease, citrate, and methyl red) or Api System^®^ (bioMérieux, Marcy l’Etoile, France).

### 2.2. Antimicrobial Susceptibility Testing

A Kirby–Bauer disk diffusion susceptibility test protocol [25] was used to determine the phenotypic antimicrobial susceptibility of *E. coli* isolates. Briefly, colonies were suspended in 5 mL of distilled, sterile water to achieve a turbidity of 0.5 on the McFarland scale. The dilution was then seeded onto Mueller–Hinton (Oxoid, UK) plates. Each isolate was tested for the following antimicrobial groups using commercial disks (Oxoid, UK): aminopenicillins (amoxicillin 25 μg and amoxicillin/clavulanic acid 20 μg/10 μg); cephalosporins (ceftiofur 30 μg, cephalexin 30 μg); carbapenems (imipenem 10 μg); quinolones (enrofloxacin 5 μg); aminoglycosides (gentamicin 10 μg, streptomycin 10 μg); macrolides (erythromycin 15 μg) tetracyclines (tetracycline 30 μg); sulfonamides (sulfamethoxazole/trimethoprim 23.75 μg/1.25 μg); florfenicol (30 μg); and lincospectin (2 μg).

Additionally, a minimum inhibitory concentration (MIC) test was performed to evaluate antimicrobial susceptibility to colistin using the broth microdilution method in 96-well plates [26]. *E. coli* ATCC 25922 was used as the quality control strain. Briefly, the tested colistin concentrations ranged from 0.25 to 8 μg/mL. The studied strains were considered resistant when their MIC value was higher than the wild-type cut-off value, which was at MIC > 2 μg/mL [27]. MIC50 and MIC90 were considered the median and percentile 90 of the population, respectively.

The susceptibility of bacteria to each antimicrobial agent was interpreted as susceptible (S), intermediate (I), and resistant (R) based on the breakpoints provided by the Clinical & Laboratory Standards Institute (CLSI) and the European Committee on Antimicrobial Susceptibility Testing (EUCAST). CLSI veterinary breakpoints were preferably used [28], and when not available, CLSI human [29], EUCAST, or CASFM veterinary breakpoints were applied as previously published by Vidal et al. [30]. In addition, multidrug resistance (MDR) was defined as resistance to at least one agent in ≥3 antimicrobial categories according to the Magiorakos et al. classification [31].

### 2.3. Statistical Analysis

Chi-squared or Fisher exact tests were used for comparison between proportions when appropriate. These statistical tests, employed to scrutinize differences in proportions, were complemented with visualizations created using R (version 4.2.1) [32] with the ggplot2 package [33]. The threshold for establishing statistical significance was set at a *p*-value of less than 0.05.

## 3. Results

### 3.1. Antimicrobial Susceptibility Testing

Antibiotic susceptibility testing was performed on 251 *E. coli* isolates: 148 were from PWD-diseased pigs and 103 were from healthy pen mates. In general, levels of AMR were high, presenting 104 out of 251 isolates (41.4%) with a multidrug resistance (MDR) profile.

Most of the *E. coli* strains presented high levels of resistance to antimicrobials commonly used in veterinary medicine. Results are presented in Table 1.

For class D antimicrobials, AMR was broadly observed to affect erythromycin (99.6%), amoxicillin (95.2%), streptomycin (91.6%), tetracycline (88.8%), lincospectin (64.5%), and sulfamethoxazole/trimethoprim (60%). For class C antimicrobials, it is interesting to note that more than 45% of the isolates showed resistance to florfenicol (Figure 1). However, of particular concern were the resistance levels within class B antimicrobials, with over half of the isolates demonstrating resistance to quinolones, particularly against enrofloxacin (56.7%). In contrast, AMR was demonstrated to be relatively lower for certain antimicrobials, such as cephalexin with 35.1% of resistance, gentamycin with 33.5%, and amoxicillin clavulanic acid with 30.7%. Additionally, the observed levels of AMR against other class B antimicrobials like colistin were notably low at 15%, and ceftiofur exhibited a resistance rate of 26.7%. Furthermore, no instances of resistance were detected for class A carbapenems, underlining their efficacy within this bacterial population (Figure 1).

### 3.2. Comparison between Diarrheic Pigs and Healthy Pen Mates

The comparison analyses between isolates from diseased and healthy pigs showed that diarrheic animals were carriers of a higher percentage of MDR bacteria compared to apparently healthy piglets (49.3% vs. 35%, *p* = 0.031). Specific differences for each antibiotic and animal group are shown in Figure 2.

In general, AMR was quite similar between both groups, with the exception of some antimicrobials such as cephalexin, ceftiofur, lincospectin, and colistin, whose resistance levels seemed more prevalent in diarrheic pigs. However, statistical differences were only observed for the cephalexin, where diarrheic animals had higher frequencies of resistance than the healthy ones (47.7% vs. 15.9%, *p* = 0.003).

Finally, given its importance in human medicine and the implementation of reduction use programs [34], colistin results from the qualitative MIC method were considered for the analysis to understand its epidemiology, as the disk diffusion method has been described as not reliable when testing polymyxin E (colistin) susceptibility. The MIC results showed that most of the isolates were susceptible to colistin (84.5%). A particular focus was put on MIC50 and MIC90 to provide a better understanding of colistin activity (Figure 3A). Thus, MIC50 was stablished at 1 μg/mL, which means that the median of *E. coli* isolates was sensible to colistin. However, MIC90 was 4 μg/mL, which, according to EUCAST, is considered a resistance profile.

Indeed, comparing diarrheic versus healthy animals, it was observed in the density plot (Figure 3B) that diarrheic animals were carriers of more resistant *E. coli* strains than their healthy pen mates. Despite the MIC50 being equal for both groups (1 μg/mL), the MIC90 was significantly different, with diarrheic animals presenting a higher concentration (4 μg/mL) than control healthy pigs (2 μg/mL) (*p* < 0.05).

## 4. Discussion

The goal of this study was to characterize the antimicrobial susceptibility pattern of 251 *E. coli* strains obtained from weaned pigs originating from farms experiencing postweaning diarrhea (PWD) and to compare the AMR patterns between pigs suffering from diarrhea and their apparently healthy pen mates.

Despite weaning being a hotspot of antimicrobial consumption, projects analyzing AMR in *E. coli* strains are usually not focused exclusively on this phase but also include other productive stages [35,36,37,38,39]. This is not criticizable at all, but given the peculiarity of weaning, a stop needs to be put there. A study from Korea [13] found high levels of AMR, very similar to those found in this study, but differed from another conducted in Australia between 1999 and 2005 [40], where levels were quite lower, showing that the extrapolation of results is not advisable in all cases.

The most remarkable findings were that 41.4% of the isolates presented an MDR profile, showing elevated resistance not only to antibiotics typically used in veterinary medicine (classified as D and C categories) but also to critical human last-resort antibiotics such as quinolones and colistin. Moreover, this high prevalence of MDR isolates, coupled with a higher MIC90 for colistin, was notably more prevalent among pigs suffering from diarrhea compared to their healthy pen mates. This discrepancy highlights a concerning correlation between disease status and the prevalence of multidrug-resistant strains, especially with higher resistance to critical antibiotics in diarrheic pigs. This would mean that the therapeutic options to treat PWD are limited. This reality should make swine practitioners aware of the need to ask for antibiotic susceptibility tests before starting antimicrobial treatments. Likewise, since weaning is the swine production phase with the highest antibiotic consumption [41], it should be convenient to overcome the antibiotic mental mark and start to fight PWD with alternatives aiming at reducing the stress derived from weaning [3], focusing on animal welfare, optimizing nutrition [42], enhancing preventive measures via cleaning and disinfection [43], employing proper vaccination protocols [44], probiotic complements [45], or other alternatives such as bacteriophages [46]. Moreover, since other pathogens, such as rotaviruses or coronaviruses, have been proven to play a role in the presentation of the PWD disease [47,48], the control of concomitant infectious diseases should be strongly emphasized along with the previously mentioned measures of biosecurity and immunological prophylaxis.

As mentioned above, the levels of resistance of *E. coli* isolates to class D antimicrobials (those recommended to use as first choice) were high in this study. Especially remarkable was the resistance to tetracycline (88.8%), which is the most used antibiotic in pig production in Spain [49]. The overuse of tetracycline could explain this high level of resistance and should make veterinarians rethink its use to deal with PWD, given its lack of efficacy. Other class D antimicrobials, such as amoxicillin (95.2%) or streptomycin (91.6%), were practically ineffective for treating these *E. coli* PWD-related strains. This finding is not surprising since *bla*EC and *bla*TEM-1B (which confer resistance to amoxicillin) and strA and strB (which make isolates resistant to streptomycin) were the most common AMR genes found in porcine *E. coli* strains [50,51].

The isolates in this study also showed relatively high resistance to class C antimicrobials. It was expected for macrolides, such as erythromycin, because the *mdf(A)* gene, which confers resistance to macrolides, is fixed on the *E. coli* chromosome [52]. However, the levels of resistance to florfenicol (45%) are worrying since it is the only phenicol licensed for the treatment of bacterial infections in food-producing animals. In contrast, other options in this category, such as amoxicillin clavulanic acid (30.7%), cephalexin (35.1%), or gentamycin (33.5%), could be better options because the levels of AMR are below 40%. However, the resistance rates are not negligible, and, again, we need to consider different options to preserve their effectiveness.

In the situation within category B, particularly concerning were the quinolones, where over half of the isolates (56.7%) exhibited resistance to enrofloxacin. This high level of resistance to quinolones raises significant concerns due to the importance of these antimicrobials in veterinary medicine. A recently published metanalysis [53] revealed Spain to have the highest recorded resistance rates, ranging between 9% and 14%. These numbers, although significant, were notably lower compared to the resistance rates uncovered in our study. This discrepancy highlights the heightened levels of resistance observed in our specific research context, underscoring the urgency for further investigation and strategic measures to address and manage antimicrobial resistance in this setting.

Resistance to other class B antimicrobials, such as third-generation cephalosporins (26.7%) and colistin (15%), was present. The dynamics of human–livestock transmission have been explored, revealing two different paradigms. In the case of ESBL bacteria (those that are resistant to third-generation cephalosporins), differences between lower- and middle-income countries (LMICs) and developed countries have been detected. While in LMICs, the bacterial interchange between human and livestock isolates is common [47], in developed countries, the transmission of these bacteria between pig farm environments and the general human population is less frequent [16,17]. However, the scenario shifts when considering colistin. The emergence of *mcr-1* in humans was notably associated with livestock, particularly swine, and aquaculture [54]. For this reason, we decided to analyze thoroughly the colistin resistance levels and calculate the minimum inhibitory concentration (MIC), revealing a concerning MIC90 of 4 μg/mL, with some isolates demonstrating even higher MIC values of 12 μg/mL. Despite the implementation of the REDUCE program, which significantly curbed colistin usage in Spain [55], our observations revealed that colistin resistance had not been eradicated. This fact could agree with Ogunlana et al. [56], who found that certain mutations in the promotor region of the mcr-1 gene could reduce its fitness cost, showing that antibiotic reduction could not be enough by itself to eliminate this specific resistance. Thus, the persistence of colistin resistance poses a challenge, indicating that despite efforts to reduce its use, resistance to this critical antibiotic remains an ongoing concern.

Finally, none of the tested *E. coli* isolates in this study displayed AMR to carbapenems, a class A antibiotic. This situation was expected because carbapenems are not used in swine production. However, despite their non-use in this context, reports of both phenotypical [57] and genotypical [58,59] resistance to these compounds in other regions worldwide require ongoing surveillance. Maintaining vigilant monitoring is crucial to track any potential emergence or spread of carbapenem resistance, despite their absence from swine production practices, and to proactively address any future challenges related to these critically important antimicrobials.

Overall, the levels of AMR were high in both pig groups, diseased and apparently healthy, suggesting that years of exposure to high antimicrobial pressure in pig farms might have contributed to this scenario. However, in our analysis of the comparison between levels of resistance among diarrheic and healthy animals, the ratio of multidrug-resistant (MDR) strains was notably higher in diseased piglets compared to their apparently healthy pen-mates. Specifically, significant differences were observed concerning cephalexin resistance, mirroring trends observed in human studies [60]. In the specific case of swine, cephalexin is commercialized via injectable solution, which ensures that only animals showing clinical symptoms are treated. In a related study focused on neonatal diarrhea within the same area [30], *E. coli* strains isolated from diarrheic animals also exhibited significantly higher resistance levels. However, the specific antimicrobials displaying resistance differed (such as quinolones and gentamicin) from our findings. Once again, these antimicrobials were predominantly available as injectable solutions. This parallel suggests a potential association warranting further investigation. However, the evidence linking the mode of antibiotic availability and resistance is not definitive and demands in-depth scrutiny for conclusive insights.

The analysis of the minimum inhibitory concentration for colistin revealed a higher MIC90 in diarrheic animals, and it is possible that this situation could be the same for other antimicrobials. It is important to note that when antimicrobials are administered through water or feed, sick animals typically reduce their consumption. Consequently, this reduction might result in lower antibiotic concentrations, leading to treatment failure as the MIC required for efficacy is not reached. This situation raises significant concerns as it suggests that the strains causing health issues might be more resilient to treatment, posing challenges to effectively addressing diseases. This discovery necessitates a comprehensive investigation, since, to our current knowledge, differences between healthy and diseased pigs during the weaning phase have not been extensively explored. Understanding these nuances in antibiotic administration and their correlation with resistance among healthy and diseased animals is pivotal for designing effective treatment strategies in pig farming contexts.

Despite this study providing a better understanding of AMR at the weaning phase, it also presents some limitations. First, this study was based on the samples submitted for diagnosis, and no representative sampling was performed, so the results are not fully extrapolated to all the territories. Second, despite the phenotype being what matters clinically, to characterize the public health risk, it is important to know the genotype, and it would be important to use other techniques such as WGS to identify the AMR responsible for the lack of effectiveness of the antibiotics. Finally, regarding the comparison between apparently healthy and diarrheic piglets, it is important to note that this condition was from a specific moment and could have changed a posteriori, but no longitudinal study was performed, so apparently healthy animals could have become diseased in the following days.

This study revealed a concerning prevalence of AMR in *E. coli* isolates, particularly in pigs suffering from diarrhea. This emphasizes that conventional antimicrobial therapy might not be as effective, highlighting the necessity of exploring alternative strategies. Implementing comprehensive measures such as stringent hygiene practices, optimizing nutritional regimens, managing stress levels, and deploying appropriate vaccination protocols becomes imperative in addressing this issue. This holistic approach aims to mitigate the reliance on antimicrobials alone, fostering a multi-faceted strategy to combat bacterial infections and AMR in pig populations.

## 5. Conclusions

This study identified high levels of AMR in PWD *E. coli* strains, with 41.4% notably being classified as multidrug-resistant strains. Moreover, it was evident that diarrheic animals harbored a greater prevalence of resistant *E. coli* strains compared to their healthy pen mates. Antimicrobials categorized under classes D and C displayed diminished effectiveness, indicating that relying solely on antibiotic treatment for PWD is inadequate. Additionally, resistance to other class B antimicrobials, such as third-generation cephalosporins and colistin, was observed, emphasizing that efforts solely focused on reducing antibiotic use may not suffice. The findings underscore the urgent need to explore new strategies and alternatives to effectively mitigate antimicrobial resistance.

## Figures and Tables

**Figure 1 animals-14-00487-f001:**
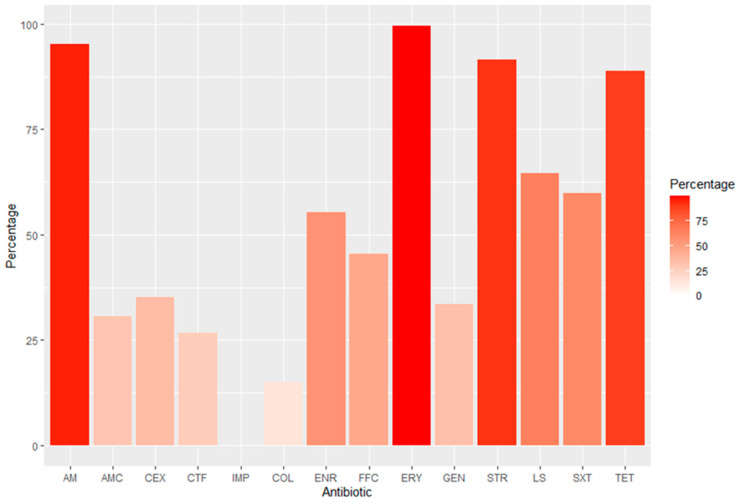
Percentage of antimicrobial resistance in *E. coli* isolates (n = 251). AM: Amoxicillin; AMC: Amoxicillin + Clavulanic Acid; CEX: Cephalexin; CTF: Ceftiofur; IMP: Imipenem; COL: Colistin; ENR: Enrofloxacin; FFC: Florfenicol; ERY: Erythromycin; GEN: Gentamycin; STR: Streptomycin; LS: Lincospectin; SXT: sulfamethoxazole/trimethoprim; TET: Tetracycline.

**Figure 2 animals-14-00487-f002:**
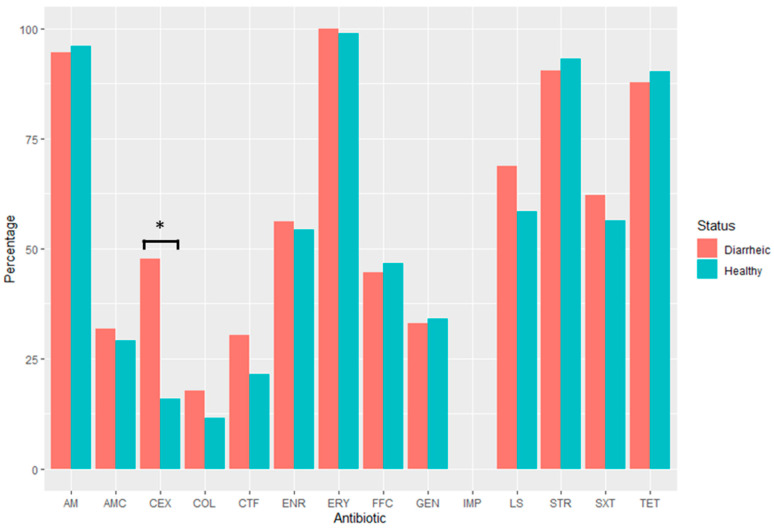
Comparison of antibiotic resistance levels between diarrheic (n = 148) and apparently healthy (n = 103) animals. AM: Amoxicillin; AMC: Amoxicillin + Clavulanic Acid; CEX: Cephalexin; CTF: Ceftiofur; IMP: Imipenem; COL: Colistin; ENR: Enrofloxacin; FFC: Florfenicol; ERY: Erythromycin; GEN: Gentamycin; STR: Streptomycin; LS: Lincospectin; SXT: sulfamethoxazole/trimethoprim; TET: Tetracycline. Red: diarrheic animals; blue: healthy animals. * *p* < 0.05.

**Figure 3 animals-14-00487-f003:**
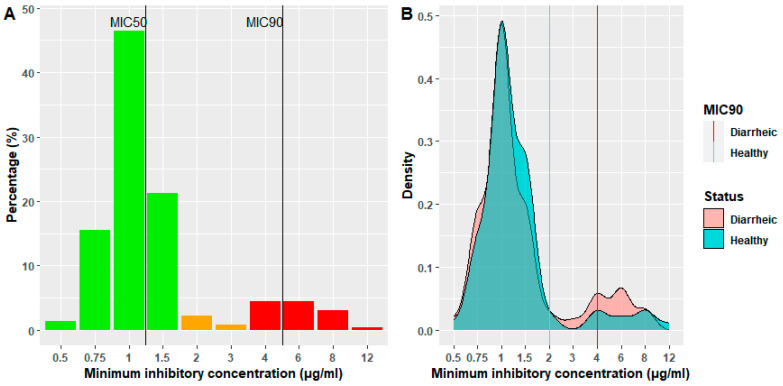
(**A**): Histogram showing the colistin minimum inhibitory concentration. Green: susceptible. Orange: intermediate. Red: resistant. The vertical lines represent percentiles 50 and 90 of minimum inhibitory concentration. (**B**): Density plot of MIC distribution comparing healthy and diarrheic animals. Red line: MIC90 of diarrheic piglets. Blue line: MIC90 of healthy piglets.

**Table 1 animals-14-00487-t001:** Comparison of resistance frequency between apparently healthy and diarrheic piglets for each antibiotic.

Class	Antimicrobial	% Resistant		
		Apparently Healthy	Diarrheic	Total
Penicillin	Amoxicillin	93.3	91.4	92.1
Penicillin + betalactamase inhibitor	Amoxicillin + clavulanic acid	29.1	31.8	31.7
1st generation cephalosporin	Cephalexin	15.9	47.8	35.1
3rd generation cephalosporin	Ceftiofur	22.2	32.6	27.7
Carbapenem	Imipenem	0.0	0.0	0.0
Polymyxins	Colistin	11.5	17.7	15.0
Quinolones	Enrofloxacin	55.5	57.5	56.7
Phenicols	Florfenicol	46.6	44.6	45.4
Macrolide	Erythromycin	99.0	100.0	99.6
Aminoglycoside	Gentamycin	34.0	33.1	33.5
Aminoglycoside	Streptomycin	93.2	90.5	91.6
Aminoglycoside	Lincospectin	59.7	66.2	63.1
Trimethoprim + sulfamide	Trimethoprim + sulphametoxazole	56.5	60.2	58.7
Tetracycline	Tetracycline	90.3	87.9	88.8

## Data Availability

Data will be made available upon reasonable request to the corresponding author.

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
