# Peer review of "Characterization of Antimicrobial Resistance in Escherichia coli Isolated from Diarrheic and Healthy Weaned Pigs in Catalonia"

_animals, 2024, doi:10.3390/ani14030487_

Round 1

Reviewer 1 Report

Comments and Suggestions for Authors

I hereby submit my comments regarding the manuscript titled “Antimicrobial resistance in Escherichia coli isolated from diarrheic and healthy weaned pigs in Catalonia’’.

I do have several comments.

Title: A word is missing in the title before antimicrobial resistance (e.g. determination of, prevalence of, characterization of)

Line 10: I would use ‘certain E. coli pathovars’’ instead of ‘’ Escherichia coli (E. coli) strains. 

Line 13: The abstract lacks a summary description of materials and methods before presenting the results.

Line 19: I would use PWD.

Line 25: What is the meaning of ‘healthy pen-mates’? I would use ‘clinically healthy piglets’. please make the change throughout the manuscript.

Line 32: ‘Dramatically’, I would change this adjective.

Line 33: The MIC unit should be corrected.

Line 49: I would use susceptible instead of prone.

Line 49-50. I would propose: the sudden change from a liquid to a solid diet leads to changes in the microbiota composition and diversity which, although not fully understood, make piglets more susceptible to PWD.

Line 61: I would propose: Could lead to the development of PWD

Line 107: antimicrobials or antibiotics ? Please standardize this usage throughout the manuscript.

Line 121: has this study been approved by the ethics committee? If yes, please add the approval number.

Lines 129-130: I was wondering why suspected E. coli isolates were identified using conventional biochemical tests or Api System® ? What biochemical tests were used? Was a PCR performed to detect a gene (e.g., the β-glucuronidase (uidA) housekeeping gene) to confirm the biochemical results?

Line 130: How many strains were identified as E. coli from the 251 fecal samples?

Line 133: At this stage after identification, are we referring to isolates or strains?

Line 135: How many isolates (n=…..) ?

Figure 1: Please added the number of isolates in the title. I would use percentage instead of “frequencies”.

Line 193: In general, AMR……

Line 196: Higher frequencies or higher prevalence?

Figure 2: Please added the number of isolates in the title for each animal category (diarrheic and clinically healthy). I would use clinically healthy animals instead of healthy animals.

Line 210: When testing bacterial susceptibility to polymyxin E (colistin).  

Line 212: Please explain in the Material and method how did you determine the MIC50 and MIC90 for colistin?

Line 2014-220: The MIC unit (mg/mL) should be corrected.

Proposal: Given that the sampling in the current study was carried out between February 2020 and December 2021, I wonder if the authors found a seasonality effect on both PWD and AMR prevalence?

Lines 237-239: Was information related to antimicrobial use collected during sampling? Were diarrheic animals treated with antibiotics? Were antibiotics used metaphylactically on all sampled farms? Were diarrheic animals separated from clinically healthy animals on the sampled farms? Can the authors classify farms according to the volume of AMU and establish the link with AMR? If this information is not available, please add it as a limitation to your study.

Lines 252-307: Please add scientific studies that have been carried out in pigs in the post-weaning period to compare with your results in terms of prevalence of AMR to each class of antibiotics? How do authors position their data in the scientific literature?

Line 317: ‘these antibiotics were predominantly available as injectable solutions”. I was wondering if quinolones and gentamicin were used in the sampled farms to support this hypothesis ?

Line 321-328: I was wondering if colistin was used in the sampled farms ?  

Line 341: Please add a section to describe the limitations of your study (representativeness of the sampled farms, the E. coli antimicrobial susceptibility analyses were carried out on a phenotypic basis only, etc.).

Line 351: I would suggest: to effectively control PWD in pigs.

Comments on the Quality of English Language

It seems to me that the English quality of the manuscript is good.

Author Response

We thank all the reviewers for their time and consideration to help us to improve our manuscript. Hereby, we have tried to address all their suggestions and concerns and we think that the result is satisfactory. We really appreciate your implication!

REVIEWER 1

I hereby submit my comments regarding the manuscript titled “Antimicrobial resistance in Escherichia coli isolated from diarrheic and healthy weaned pigs in Catalonia’’.

 I do have several comments.

Title: A word is missing in the title before antimicrobial resistance (e.g. determination of, prevalence of, characterization of)

Answer: thanks for the comment. We have modified the title as follows: Characterization of antimicrobial resistance in Escherichia coli isolated from diarrheic and healthy weaned pigs in Catalonia.

Line 10: I would use ‘certain E. coli pathovars’’ instead of ‘’ Escherichia coli (E. coli) strains.

Answer: we agree with the change.

Line 13: The abstract lacks a summary description of materials and methods before presenting the results.

Answer: we have included this new sentence describing M&M: “Herein, we tested the susceptibility to fourteen antimicrobials in 251 E. coli strains isolated from fecal samples of diarrheic (n=148) and apparently healthy piglets (n=103) in farms of Catalonia.”

Line 19: I would use PWD.

Answer: changed in line 22.

Line 25: What is the meaning of ‘healthy pen-mates’? I would use ‘clinically healthy piglets’. please make the change throughout the manuscript.

Answer: We think that probably “apparently healthy” would fit better since we cannot ensure that the animals would not develop disease in the following days. Thus, we have reworded the healthy pen mates as apparently healthy pen mates throughout the manuscript.

Line 32: ‘Dramatically’, I would change this adjective.

Answer: Dramatically has been changed by substantially

Line 33: The MIC unit should be corrected.

Answer: corrected in line 38

Line 49: I would use susceptible instead of prone.

Answer: corrected

Line 49-50. I would propose: the sudden change from a liquid to a solid diet leads to changes in the microbiota composition and diversity which, although not fully understood, make piglets more susceptible to PWD.

Answer: we agree with the suggestion and the modification has been done.

Line 61: I would propose: Could lead to the development of PWD

Answer: Done in line 71

Line 107: antimicrobials or antibiotics ? Please standardize this usage throughout the manuscript.

Answer: We have standardized the word “antimicrobials” throughout the manuscript.

Line 121: has this study been approved by the ethics committee? If yes, please add the approval number.

Answer: The samples were taken by field veterinarians from clinical cases and submitted to our lab for diagnosis, so approval by any  ethics committee was not necessary.

Lines 129-130: I was wondering why suspected E. coli isolates were identified using conventional biochemical tests or Api System® ? What biochemical tests were used? Was a PCR performed to detect a gene (e.g., the β-glucuronidase (uidA) housekeeping gene) to confirm the biochemical results?

Answer: Biochemical tests (oxidase, catalase, TSI, SIM, urease, citrate, and methyl red) were used initially to identify E. coli strains. If the identification was not reliable, Api System was used.

Line 130: How many strains were identified as E. coli from the 251 fecal samples?

Answer: in diseased pigs all cultures were pure isolates and from apparently healthy ones we select the most predominant growth. Thus, we identified 251 strains (one from each animal).

Line 133: At this stage after identification, are we referring to isolates or strains?

Answer: Isolates

Line 135: How many isolates (n=…..) ?

Answer: 251 isolates (added in line 152)

Figure 1: Please added the number of isolates in the title. I would use percentage instead of “frequencies”.

Answer: Done in line 213

Line 193: In general, AMR……

Answer: Done

Line 196: Higher frequencies or higher prevalence?

Answer: Prevalence is a measure of disease frequency. In this case we are comparing diarrheic and healthy animals, so frequency seems to fit better.

Figure 2: Please added the number of isolates in the title for each animal category (diarrheic and clinically healthy). I would use clinically healthy animals instead of healthy animals.

Answer: We have changed it by apparently healthy

Line 210: When testing bacterial susceptibility to polymyxin E (colistin). 

Answer: we agree with the comment.

Line 212: Please explain in the Material and method how did you determine the MIC50 and MIC90 for colistin?

Answer: lines 167-168: we have introduced this new sentence: “MIC50 and MIC90 were considered as the median and percentile 90 of the population respectively”.

Line 2014-220: The MIC unit (mg/mL) should be corrected.

Answer: corrected

Proposal: Given that the sampling in the current study was carried out between February 2020 and December 2021, I wonder if the authors found a seasonality effect on both PWD and AMR prevalence?

Answer: It was already studied but we did not find statistical differences

Lines 237-239: Was information related to antimicrobial use collected during sampling? Were diarrheic animals treated with antibiotics? Were antibiotics used metaphylactically on all sampled farms? Were diarrheic animals separated from clinically healthy animals on the sampled farms? Can the authors classify farms according to the volume of AMU and establish the link with AMR? If this information is not available, please add it as a limitation to your study.

Answer: We know that, at the moment of sampling (postweaning) animals were not receiving antibiotic treatment. The information that we lack is the information about the previous history of treatments that could be influencing the AMR levels. This is now included as a limitation. Clinically healthy animals were located in the same pen that diarrheic ones.

Lines 252-307: Please add scientific studies that have been carried out in pigs in the post-weaning period to compare with your results in terms of prevalence of AMR to each class of antibiotics? How do authors position their data in the scientific literature?

Answer: Initially it was not stated, since very few studies are exclusively based on weaning phase (which has specific characteristics such high antibiotic use or stressed animals), with data from other phases confusing the results. Now, we have included the data from two studies only focused there.

Line 317: ‘these antibiotics were predominantly available as injectable solutions”. I was wondering if quinolones and gentamicin were used in the sampled farms to support this hypothesis ?

Answer: All we know is that, at the moment of sampling, no antibiotic treatment was being administered. However, we do not know the previous history of treatments.

Line 321-328: I was wondering if colistin was used in the sampled farms? 

Answer: To the best of our knowledge, it was not used. However, farmers could have been used it without informing the veterinarian, but, at least, it was not reported. However, in recent years, the colistin use has been almost eliminated in Spain, so it would be rare that these farms were using it.

Line 341: Please add a section to describe the limitations of your study (representativeness of the sampled farms, the E. coli antimicrobial susceptibility analyses were carried out on a phenotypic basis only, etc.).

Answer: We have added a paragraph with the possible limitation to extrapolate the results of the study (lines 378-388).

Line 351: I would suggest: to effectively control PWD in pigs.

Answer: Done

Reviewer 2 Report

Comments and Suggestions for Authors

This manuscript aims to investigate antimicrobial susceptibility phenotype patterns in E. coli isolated from farms experiencing postweaning diarrhea (PWD) in Catalonia, and also to investigate, whether there are distinct antimicrobial susceptibility phenotype patterns in E. coli strains associated with PWD from diarrheic piglets compared to those found in healthy animals within the same environment.

As referred by authors, field studies are important to create adequate guidelines to help veterinary practitioners in using antimicrobials in ways that prioritize animal health while minimizing the risk of antimicrobial resistance development.

The following revisions are suggested:

Acronyms and abbreviations should be defined the first time they appear in the text: Line 152 – Comité de l’Antibiogramme de la Société Française de Microbiologie (CASFM

Line 213, 218 – “1μg/ml” instead of “1μ/ml”

Line 214, 220, 291 – “4μg/ml” instead of “4μ/ml”

Line 217 – put “E. coli” in italics

Line 220 - “2μg/ml” instead of “2μ/ml”

Line 292 - “12μg/ml” instead of “12μ/ml”

References 50, 51, 52, 53 and 54 must be included in the references list.

Comments on the Quality of English Language

We suggest:

Line 274 - “A recently published metanalysis…” instead of “In a recently published metanalysis…”

Line 280 - “...class B antibiotics such as third generation …” instead of “...class B antibiotics such third generation..”

Author Response

We thank all the reviewers for their time and consideration to help us to improve our manuscript. Hereby, we have tried to address all their suggestions and concerns and we think that the result is satisfactory. We really appreciate your implication.

The following revisions are suggested:

Acronyms and abbreviations should be defined the first time they appear in the text: Line 152 – Comité de l’Antibiogramme de la Société Française de Microbiologie (CASFM

Line 213, 218 – “1μg/ml” instead of “1μ/ml”

Line 214, 220, 291 – “4μg/ml” instead of “4μ/ml”

Line 217 – put “E. coli” in italics

Line 220 - “2μg/ml” instead of “2μ/ml”

Line 292 - “12μg/ml” instead of “12μ/ml”

Answer: all these errors have been corrected in the text.

References 50, 51, 52, 53 and 54 must be included in the references list.

Answer: we have double-check the references list and now all references are included.

Line 274 - “A recently published metanalysis…” instead of “In a recently published metanalysis…”

Line 280 - “...class B antibiotics such as third generation …” instead of “...class B antibiotics such third generation.”

Answer: changes made.

Reviewer 3 Report

Comments and Suggestions for Authors

Dear Authors and Editor,

I appreciate review the paper “Antimicrobial resistance in Escherichia coli isolated from diarrheic and healthy weaned pigs in Catalonia”.

The aim of this paper is very important to swine production. 

General comments: 

- it is necessary review the italic in E. coli and in genes names (discussion);

- it is better to standardize when writing about enrofloxacin class (quinolone or fluoroquinolone);

- in introduction, the authors must stick to the main topic. Lines 57 to 66, for example, could be delete. After writing the E. coli agent and its importance to PWD, explain about antimicrobial resistance. It is important to highlight antimicrobial resistance being important for public health (how the authors explain), and it is a big problem for pig farming, limiting the antibiotics available for animal treatments;

- in the results, the authors did not mention about PDR (pan drug) samples. As mentioned in the material and methods, it is important to mention in the results;

- authors can also show the antimicrobial results (frequency) in a table, separating classes, like this:

Class    Antimicrobial                         Health             Diarrheic         Total

                                                      % (resistant)

- in discuss it is important to mention and discuss about PDR samples (or delete in material and methods); 

- in discuss, it is necessary explain more about the high levels of resistance found in healthy animals. Despite showing a difference in MDR between diarrheic and healthy animals, healthy animals also showed high levels of resistance. Why? It would be interesting to discuss this further and not just in lines 308-315.

Author Response

We thank all the reviewers for their time and consideration to help us to improve our manuscript. Hereby, we have tried to address all their suggestions and concerns and we think that the result is satisfactory. We really appreciate your implication

General comments:

- it is necessary review the italic in E. coli and in genes names (discussion); - it is better to standardize when writing about enrofloxacin class (quinolone or fluoroquinolone);

Answer: we have corrected these issues in the revised version.

- in introduction, the authors must stick to the main topic. Lines 57 to 66, for example, could be delete. After writing the E. coli agent and its importance to PWD, explain about antimicrobial resistance. It is important to highlight antimicrobial resistance being important for public health (how the authors explain), and it is a big problem for pig farming, limiting the antibiotics available for animal treatments;

Answer: we have changed the introduction accordingly in lines 83-85.

- in the results, the authors did not mention about PDR (pan drug) samples. As mentioned in the material and methods, it is important to mention in the results;

Answer: Since any sample was PDR, we have delete it from Material and Methods

- authors can also show the antimicrobial results (frequency) in a table, separating classes, like this:

Class    Antimicrobial                         Health             Diarrheic         Total

                                                      % (resistant)

Answer: Table 1 has been modified accordingly.

- in discuss it is important to mention and discuss about PDR samples (or delete in material and methods);

Answer: as mentioned above, we have deleted it from Material and Methods

- in discuss, it is necessary explain more about the high levels of resistance found in healthy animals. Despite showing a difference in MDR between diarrheic and healthy animals, healthy animals also showed high levels of resistance. Why? It would be interesting to discuss this further and not just in lines 308-315.

Answer: we have included a paragraph according to this observation (lines 349-351).

Reviewer 4 Report

Comments and Suggestions for Authors

Dear Editor

In this paper, a comparison of the E. coli isolates recovered from both deceased and healthy piglets is being made. Generally the presentation of the methods used and the results obtained is adequate. However the experimental procedure reported lacks specific parameters that are important for the comparison of these strains; 1) no history of disease or antibiotic usage has been collected prior to sampling 2) only phenotypic examination of antimicrobial resistance was performed 3) no categorisation of the strains in clusters or pathogenicity groups has been made.

P2 L55. eae is the gene encoding intimin. Please revise.

P2 L74-6. Please rephrase.

P3 L115. Materials and Methods. Since a comparison between isolates from deceased and healthy animals is being made, it is of paramount importance to collect information concerning the use of antibiotics to these animals prior to death. Any relevant information is missing.

P3 L115. Materials and Methods. No identification of the genetic background of antimicrobial resistance has been carried out. The phenotype is what matters most; still, in order to compare strains, it is desirable to examine for the genetic background of the exhibited resistance, especially since it is reported in the Discussion part (P8 L259, P8 L288, etc.). Further, the strains should have been allocated to pathogenicity groups since the importance of E. coli to public and swine health can vary for harmless commensals to highly pathogenic strains.

P3 L136. The producer of the disks is required.

P4 L164-182. It is advised to add the confidence intervals of the resistance rates along with the values.

P6 L206-7. A reference is needed.

P7 L255-7. The density plot is quite fancy but does not add to the interpretation of results. The histogram reporting MICs for both deceased and healthy animal is adequate.

P7 L277. The Discussion part is mostly an enhanced report of the results or generic extrapolation of remarks, with limited comparison to the results or conclusions of other researchers.

Comments on the Quality of English Language

English language is mostly fine; some minor edits spotted can be corrected further on.

Author Response

We thank all the reviewers for their time and consideration to help us to improve our manuscript. Hereby, we have tried to address all their suggestions and concerns and we think that the result is satisfactory. We really appreciate your implication.

P2 L55. eae is the gene encoding intimin. Please revise.

P2 L74-6. Please rephrase.

Answer: changes done

P3 L115. Materials and Methods. Since a comparison between isolates from deceased and healthy animals is being made, it is of paramount importance to collect information concerning the use of antibiotics to these animals prior to death. Any relevant information is missing.

Answer: All we know is that, at the day of sampling, no antibiotic treatment was being administered. However, we do not know the previous history of treatments. This information has been included as a limitation in the discussion section (lines 380-390).

P3 L115. Materials and Methods. No identification of the genetic background of antimicrobial resistance has been carried out. The phenotype is what matters most; still, in order to compare strains, it is desirable to examine for the genetic background of the exhibited resistance, especially since it is reported in the Discussion part (P8 L259, P8 L288, etc.). Further, the strains should have been allocated to pathogenicity groups since the importance of E. coli to public and swine health can vary for harmless commensals to highly pathogenic strains.

Answer: we agree with this observation, but the objective of this study was focused on the resistance phenotype. However, we have included that genotyping was also important to determine the AMR genes, as a limitation of this study in the discussion section (lines 380-390)

P3 L136. The producer of the disks is required.

Answer: Done

P4 L164-182. It is advised to add the confidence intervals of the resistance rates along with the values.

P6 L206-7. A reference is needed.

Answer: Done

P7 L255-7. The density plot is quite fancy but does not add to the interpretation of results. The histogram reporting MICs for both deceased and healthy animal is adequate.

Answer: We thank the comment. First, we tried to visualize it in the suggested way, but it did not convince us since it was not very clear, and we thought that the density plot showed clearer what we wanted to show. In any case, if the editor considers necessary to change it we can reconsider it.

P7 L277. The Discussion part is mostly an enhanced report of the results or generic extrapolation of remarks, with limited comparison to the results or conclusions of other researchers.

Answer: We agree with the comment. We have introduced a new paragraph (lines 261-267) concerning this issue.